# Future-Dependent Value-Based
# Off-Policy Evaluation in POMDPs

**Masatoshi Uehara** [*]
Genentech
uehara.masatoshi@gene.com

**Haruka Kiyohara** [†]
Cornell University
hk844@cornell.edu

**Andrew Bennett**
Morgan Stanley [‡]
Andrew.Bennett@morganstanley.com

**Victor Chernozhukov**
MIT
vchern@mit.edu

**Nan Jiang**
UIUC
nanjiang@illinois.edu

**Nathan Kallus**
Cornell University
kallus@cornell.edu

**Chengchun Shi**
LSE
c.shi7@lse.ac.uk

**Wen Sun**
Cornell University
ws455@cornell.edu

## Abstract

We study off-policy evaluation (OPE) for partially observable MDPs (POMDPs) with general function approximation. Existing methods such as sequential importance sampling estimators suffer from the curse of horizon in POMDPs. To circumvent this problem, we develop a novel model-free OPE method by introducing future-dependent value functions that take future proxies as inputs and perform a similar role to that of classical value functions in fully-observable MDPs. We derive a new off-policy Bellman equation for future-dependent value functions as conditional moment equations that use history proxies as instrumental variables. We further propose a minimax learning method to learn future-dependent value functions using the new Bellman equation. We obtain the PAC result, which implies our OPE estimator is close to the true policy value under Bellman completeness, as long as futures and histories contain sufficient information about latent states. Our code is available at https://github.com/aiueola/neurips2023-future-dependent-ope.

## 1 Introduction

Reinforcement learning (RL) has demonstrated success when it is possible to interact with the environment and collect data in an adaptive manner. However, for domains such as healthcare, education, robotics, and social sciences, online learning can be risky and unethical [e.g., KHS[+]15, KL19, TDHL19, SRRH21]. To address this issue, a variety of offline RL methods have recently been developed for policy learning and evaluation using historical data in these domains [see LKTF20, for an overview]. In this paper, we focus on the off-policy evaluation (OPE) problem, which concerns estimating the value of a new policy (called the evaluation policy) using offline log data that was

---

[*]Co-first author. This work was done at Cornell University

[†]Co-first author. This work was done at Tokyo Institute of Technology

[‡]This work was done at Cornell University

37th Conference on Neural Information Processing Systems (NeurIPS 2023).

generated under a different policy (called the behavior policy) [TB16, JL16, TDHL19, CQ22]. OPE is especially useful in the aforementioned high-stakes domains.

We focus on off-policy evaluation (OPE) in partially observable Markov decision processes (POMDPs). Partial observability is a common phenomenon in practical applications [Cas98, KLC98, XLSS20], and it poses a serious challenge for sample-efficient learning. Most existing OPE methods are developed for MDPs and their statistical learning guarantees rely crucially on the Markov assumption. To extend these methods to POMDPs, one may use the entire history of observations as a *state* to satisfy Markovanity. This allows us to apply sequential importance sampling [SIS, Pre00, JSLZ19, HW21] or its variant [e.g., sequential doubly robust (SDR) methods RRS98, MvdLRG01, JL16, TB16, FCG18, BMVVDL19] to the transformed data for valid policy evaluation. An alternative approach is to employ value-based methods by constructing a history-dependent Q-function that takes the entire history as input. These value functions can be estimated via fitted-Q evaluation [FQE, EGW05, MS08, LVY19] or minimax methods [ASM08, CJ19, FLL19, NCDL19, UHJ20, UIJ$^+$21, ZW22]. However, all aforementioned estimators suffer from the curse of horizon, as their estimation errors grow exponentially with respect to the (effective) horizon and become prohibitively large in long-horizon settings [LLTZ18, KU20, KU22].

The goal of this paper is to devise practical and efficient OPE methods in large partially observable environments, while breaking the curse of horizon. As a first step, we restrict our attention to the evaluation of short memory-based policies that take several recent observations instead of the entire history as inputs. Short memory-based policies are widely used in practice, since maintaining the whole previous observation leads to a well-known computational challenge dubbed as the curse of history [PGT06, GMR22]. For example, in the deep Q-network (DQN) algorithm, four game frames are stacked together to train the optimal policy in Atari [MKS$^+$13], while Open AI Five uses a window of length 16 [BBC$^+$19]. Even when considering the evaluation of policies with short-memory or memory-less characteristics (i.e., policies that depend solely on current observations), naïve methods such as SIS, SDR, and FQE still suffer from the curse of horizon[4].

Our proposal takes a model-free perspective and introduces a new concept called "future-dependent value functions". The proposed value function does not involve latent states which are unobservable in POMDPs. Nor does it relies on the entire data history. It performs the role of standard value functions by incorporating future observations as proxies for latent states. We demonstrate that these future-dependent value functions satisfy a new off-policy Bellman equation for POMDPs, which involves past observations to approximate latent states. To estimate these future-dependent value functions from offline data using the new Bellman equation, we propose minimax learning methods that accommodate various function approximation, including deep neural networks, RKHS, and linear models. Notably, our method extends the classical LSTD [LP03] developed in MDPs to the POMDP setting when using linear models. To summarize, our contributions are as follows:

- We propose a novel model-free method for OPE that leverages future-dependent value functions. Our key idea is to utilize future and historical observations as proxies for latent states.

- We derive a new Bellman equation for learning future-dependent value functions. The proposed estimator can accommodate various types of function approximations.

- We provide PAC guarantees to demonstrate that our method can address the curse of horizon, and conduct numerical experiments to showcase its superiority over existing methods.

Note that similar concepts of future-dependent value functions have recently been introduced in the context of OPE in confounded POMDPs [SUHJ22] and online RL in POMDPs [USL$^+$22]. However, we focus on OPE in *non-confounded* POMDPs. Further detailed discussions can be found in Section 1.1.

## 1.1 Related Literature

**OPE in confounded POMDPs.** OPE with unmeasured confounders has been actively studied [ZB16, NKYB20, TSM20, WYW20, KZ20, LFY$^+$21, NJ21, GCZ$^+$22, SZY$^+$22, XZS$^+$22, LMWY22, CWWY23, BSZ23]. Among them, [TSM20] adopted the POMDP model to formulate the OPE problem in the presence of unmeasured confounders. They borrowed ideas from the causal inference literature on double negative control adjustment [MGT18, MST18, CPS$^+$20,

---

[4]Notice that even when restricting to short-memory policies, it is necessary to use the entire history instead of the short memory as the state to meet the Markov property.

KMU21, SMNTT20] to derive consistent value estimators in tabular settings. Later, [BKLM21] and [SUHJ22] extend their proposal by incorporating confounding bridge functions, thereby enabling general function approximation. (see the difference between these bridge functions and the proposed future-dependent value function in Section B). However, these methods do not apply to our unconfounded POMDP setting. This is because these methods require the behavior policy to *only* depend on the latent state to ensure certain conditional independence assumptions. These assumptions are violated in our setting, where the behavior policy may depend on the observation – a common scenario in practical applications. In addition, we show that in the unconfounded setting, it is feasible to leverage multi-step future observations to relax specific rank conditions in their proposal, which is found to be difficult in the confounded setting [NJ21]. Refer to Example 1.

**Online learning in POMDPs.** In the literature, statistically efficient online learning algorithms with polynomial sample complexity have been proposed in tabular POMDPs [GDB16, ALA16, JKKL20, LCSJ22], linear quadratic Gaussian setting (LQG) [LAHA21, SSH20], latent POMDPs [KECM21] and reactive POMDPs/PSRs [KAL16, JKA+17]. All the methods above require certain model assumptions. To provide a more unified framework, researchers have actively investigated online learning in POMDPs with general function approximation, as evidenced by a vast body of work [ZUSL22, LNSJ22, CBM22, ZXZ+22]. As the most relevant work, by leveraging future-dependent value functions, [USL+22] propose an efficient PAC RL algorithm in the online setting. They require the existence of future-dependent value functions and a low-rank property of the Bellman loss. Instead, in our approach, while we similarly require the existence of future-dependent value functions, we do not require the low-rank property of Bellman loss. Instead, we use the invertibility condition in Theorem 1, i.e., we have informative history proxies as instrumental variables. As a result, the strategies to ensure efficient PAC RL in POMDPs are quite different in the offline and online settings.

**Learning dynamical systems via spectral learning.** There is a rich literature on POMDPs by representing them as predictive state representations (PSRs) [Jae00, LS01, RGT04, SJR04]. PSRs are models of dynamical systems that represent the state as a vector of predictions about future observable events. They are appealing because they can be defined directly from observable data without inferring hidden variables and are more expressive than Hidden Markov Models (HMMs) and POMDPs [SJR04]. Several spectral learning algorithms have been proposed for PSRs [BSG11, HKZ12, BGG13], utilizing conditional mean embeddings or Hilbert space embeddings. These approaches provide closed-form learning solutions, which simplifies computation compared to EM-type approaches prone to local optima and non-singularity issues. While these methods have demonstrated success in real-world applications [BSG11] and have seen subsequent improvements [KJS15, HDG15, VSH+16, DHB+17], it is still unclear how to incorporate more flexible function approximations such as neural networks in a model-free end-to-end manner with a valid PAC guarantee. Our proposed approach not only offers a new model-free method for OPE, but it also incorporates these model-based spectral learning methods when specialized to dynamical system learning with linear models (see Section F).

**Planning in POMDPs.** There is a large amount of literature on planning in POMDPs. Even if the models are known, exact (or nearly optimal) planning in POMDPs is known to be NP-hard in the sense that it requires exponential time complexity with respect to the horizon [PT87, BDRS96]. This computational challenge is often referred to as the *curse of history* [PGT06]. A natural idea to mitigate this issue is to restrict our attention to short-memory policies [AYA18, MY20, KY22, GMR22]. Practically, a short-memory policy can achieve good empirical performance, as mentioned earlier. This motivates us to focus on evaluating short-memory policies in this paper.

## 2 Preliminaries

In this section, we introduce the model setup, describe the offline data and present some notations.

**Model setup.** We consider an infinite-horizon discounted POMDP $\mathcal{M} = \langle \mathcal{S}, \mathcal{A}, \mathcal{O}, r, \gamma, \mathbb{O}, \mathbb{T} \rangle$ where $\mathcal{S}$ denotes the state space, $\mathcal{A}$ denotes the action space, $\mathcal{O}$ denotes the observation space, $\mathbb{O} : \mathcal{S} \to \Delta(\mathcal{O})$ denotes the emission kernel (i.e., the conditional distribution of the observation given the state), $\mathbb{T} : \mathcal{S} \times \mathcal{A} \to \Delta(\mathcal{S})$ denotes the state transition kernel (i.e., the conditional distribution of the next state given the current state-action pair), $r : \mathcal{S} \times \mathcal{A} \to \mathbb{R}$ denotes the reward function, and $\gamma \in [0, 1)$ is the discount factor. All the three functions $\mathbb{T}, r, \mathbb{O}$ are unknown to the learner.

For simplicity, we consider the evaluation of memory-less policies $\pi : \mathcal{O} \to \Delta(\mathcal{A})$ that solely depend on the current observation $O$ in the main text. The extension to policies with short-memory is discussed in Section C.

Next, given a memory-less evaluation policy $\pi^e$, we define the parameter of interest, i.e., the policy value. Following a policy $\pi^e$, the data generating process can be described as follows. First, $S_0$ is generated according to some initial distribution $\nu_S \in \Delta(\mathcal{S})$. Next, the agent observes $O_0 \sim \mathbb{O}(\cdot \mid S_0)$, executes the initial action $A_0 \sim \pi^e(\cdot \mid O_0)$, receives a reward $r(S_0, A_0)$, the environment transits to the next state $S_1 \sim \mathbb{T}(\cdot \mid S_0, A_0)$, and this process repeats. Our objective is to estimate

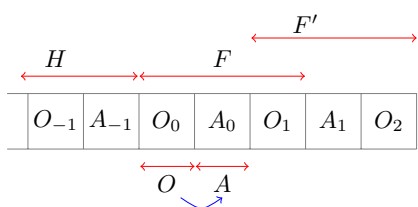

Figure 1: Case with $M_H = 1, M_F = 2$. An action $A$ is generated depending on $O$. The extension to memory-based policy is discussed in Section C.

$$J(\pi^e) := \mathbb{E}_{\pi^e}\left[\sum_{t=0}^{\infty} \gamma^t R_t\right],$$

where the expectation $\mathbb{E}_{\pi^e}$ is taken by assuming the data trajectory follows the evaluation policy $\pi^e$.

**Offline data.**    We convert the trajectory data generated by a behavior policy $\pi^b : \mathcal{O} \to \Delta(\mathcal{A})$, into a set of history-observation-action-reward-future transition tuples (denoted by $\mathcal{D}_{\text{tra}}$) and a set of initial observations (denoted by $\mathcal{D}_{\text{ini}}$). The first data subset enables us to learn the reward, emission and transition kernels whereas the second data subset allows us to learn the initial observation distribution.

Specifically, the dataset $\mathcal{D}_{\text{tra}}$ consists of $N$ data tuples $\{(H^{(i)}, O^{(i)}, A^{(i)}, R^{(i)}, F'^{(i)})\}_{i=1}^N$. We use $(H, O, A, R, F')$ to denote a generic history-observation-action-reward-future tuple where $H$ denotes the $M_H$-step historical observations obtained prior to the observation $O$ and $F'$ denotes the $M_F$-step future observations after $(O, A)$ for some integers $M_H, M_F \geq 1$. Specifically, at a given time step $t$ in the data trajectory, we use $(O, A, R)$ to denote $(O_t, A_t, R_t)$, and set

$$H = (O_{t-M_H:t-1}, A_{t-M_H:t-1}) \text{ and } F' = (O_{t+1:t+M_F}, A_{t+1:t+M_F-1}).$$

We additionally set $F = (O_{t:t+M_F-1}, A_{t:t+M_F-2})$. Note we use the prime symbol ' to represent the next time step. See Figure 1 for details when we set $t = 0$.

Throughout this paper, we use uppercase letters such as $(H, S, O, A, R, S', F')$ to denote *random variables* in the offline data, and lowercase letters such as $(h, s, o, a, r, s', f')$ to denote their *realizations*, unless stated otherwise. To simplify the presentation, we assume the stationarity of the environment, i.e., the marginal distributions of $(H, S, F)$ and $(H', S', F')$ are identical.

The dataset $\mathcal{D}_{\text{ini}}$ consists of $N'$ data tuples $\{O_{0:M_F-1}^{(i)}, A_{0:M_F-2}^{(i)}\}_{i=1}^{N'}$ which is generated as follows: $S_0 \sim \nu_S, O_0 \sim \mathbb{O}(\cdot \mid S_0), A_0 \sim \pi^b(\cdot \mid O_0), S_1 \sim \mathbb{T}(\cdot \mid S_0, A_0), \cdots$, until we observe $O_{M_F-1}^{(i)}$ and $A_{M_F-1}^{(i)}$. We denote its distribution over $\mathcal{F} = \mathcal{O}^{M_F} \times \mathcal{A}^{M_F-1}$ by $\nu_{\mathcal{F}}(\cdot)$.

**Remark 1** (Standard MDPs). *Consider the setting where $S = O$ and $M_H = 0, M_F = 1$. In that case, we set $H$ to $S$ instead of histories. Then, $\mathcal{D}_{\text{tra}} = \{O^{(i)}, A^{(i)}, R^{(i)}, O'^{(i)}\}$. We often assume $\nu_{\mathcal{O}}$ ($\nu_{\mathcal{F}}$ in our setting) is known. This yields the standard OPE setting in MDPs [CJ19, UHJ20].*

**Notation.**    We streamline the notation as follows. We define a state value function under $\pi^e$: $V^{\pi^e}(s) := \mathbb{E}_{\pi^e}[\sum_{k=0}^{\infty} \gamma^k R_k \mid S_0 = s]$ for any $s \in \mathcal{S}$. Let $d_t^{\pi^e}(\cdot)$ be the marginal distribution of $S_t$ under the policy $\pi^e$. Then, we define the discounted occupancy distribution $d_{\pi^e}(\cdot) := (1 - \gamma)\sum_{t=0}^{\infty} \gamma^t d_t^{\pi^e}(\cdot)$. We denote the domain of $F, H$ by $\mathcal{F} = (\mathcal{O} \times \mathcal{A})^{M_F-1} \times \mathcal{O}, \mathcal{H} = (\mathcal{O} \times \mathcal{A})^{M_H}$, respectively. The notations $\mathbb{E}, \mathbb{E}_{\mathcal{D}}$ (without any subscripts) represent the population or sample average over the offline data $\mathcal{D} = \mathcal{D}_{\text{tra}} \cup \mathcal{D}_{\text{ini}}$, respectively. We denote the distribution of offline data by $P_{\pi^b}(\cdot)$. We define the marginal density ratio $w_{\pi^e}(S) := d_{\pi^e}(S)/P_{\pi^b}(S)$. Given a matrix $C$, we denote its Moore-Penrose inverse by $C^+$ and its smallest singular value by $\sigma_{\min}(C)$. For a given integer $m > 0$, let $I_m$ denote an $m \times m$ identity matrix. Let $\otimes$ denote outer product and $[T] = \{0, \cdots, T\}$ for any integer $T > 0$.

## 3  Identification via Future-dependent Value Functions

In this section, we present our proposal to identify policy values under partial observability by introducing future-dependent value functions. We remark that the target policy's value is identifiable from the observed data via SIS or SDR. Nonetheless, as commented earlier in Section 1, these methods suffer from the curse of horizon. Here, we propose an alternative identification approach that

can possibly circumvent the curse of horizon. It serves as a building block to motivate the proposed estimation methods in Section 4.

In fully observable MDPs, estimating a policy's value is straightforward using the value function-based method $J(\pi^e) = \mathbb{E}_{s \sim \nu_S}[V^{\pi^e}(s)]$. However, in partial observable environments where the latent state is inaccessible, the state value function $V^{\pi^e}(s)$ is unidentifiable. To address this challenge, we propose the use of *future-dependent value functions* that are defined based on observed variables and serve a similar purpose to state value functions in MDPs.

**Definition 1** (Future-dependent value functions). *Future-dependent value functions $g_V \in [\mathcal{F} \to \mathbb{R}]$ are defined such that the following holds almost surely,*

$$\mathbb{E}[g_V(F) \mid S] = V^{\pi^e}(S).$$

*Recall that the expectation is taken with respect to the offline data generated by $\pi^b$.*

Crucially, the future-dependent value functions mentioned above may not always exist, and they need not be unique. Existence is a vital assumption in our framework, although we don't insist on uniqueness. We will return to the topics of existence and uniqueness after demonstrating their relevance in offline policy evaluation.

Hereafter, we explain the usefulness of future-dependent value functions. Future-dependent value functions are defined as embeddings of value functions on latent states onto multi-step futures. Notice that $J(\pi^e) = \mathbb{E}_{s \sim \nu_S}[V^{\pi^e}(s)] = \mathbb{E}_{f \sim \nu_{\mathcal{F}}}[g_V(f)]$. These future-dependent value functions are useful in the context of OPE as they enable us to accurately estimate the final policy value. However, the future-dependent value function itself cannot be identified since its definition relies on unobserved states. To overcome this challenge, we introduce a learnable counterpart called the *learnable future-dependent value function*. This learnable version is defined based on observed quantities and thus can be identified.

**Definition 2** (Learnable future-dependent value functions). *Define $\mu(O, A) := \pi^e(A \mid O)/\pi^b(A \mid O)$. Learnable future-dependent value functions $b_V \in [\mathcal{F} \to \mathbb{R}]$ are defined such that the following holds almost surely,*

$$0 = \mathbb{E}\left[\mu(O, A)\{R + \gamma b_V(F')\} - b_V(F) \mid H\right]. \tag{1}$$

*Recall that the expectation is taken with respect to the offline data generated by $\pi^b$. We denote the set of solutions by $\mathcal{B}_V$.*

To motivate this definition, we recall that the off-policy Bellman equation in MDPs [DNP+14] can be expressed as follows:

$$V^{\pi^e}(S) = \mathbb{E}\left[\mu(O, A)(R + \gamma V^{\pi^e}(S')) \mid S\right]. \tag{2}$$

Then, by the definition of future-dependent value functions and certain conditional independence relations $(F' \perp (O, A) \mid S')$, we obtain that

$$0 = \mathbb{E}\left[\mu(O, A)\{R + \gamma g_V(F')\} - g_V(F) \mid S\right]. \tag{3}$$

Since $H \perp (O, A, F') \mid S$, taking another conditional expectation given $H$ on both sides yields that

$$0 = \mathbb{E}\left[\mu(O, A)\{R + \gamma g_V(F')\} - g_V(F) \mid H\right]. \tag{4}$$

Therefore, (4) can be seen as an off-policy Bellman equation for future-dependent value functions, analogous to the Bellman equation (2) in MDPs. Based on the above discussion, we present the following lemma.

**Lemma 1.** *Future-dependent value functions are learnable future-dependent value functions.*

**Remark 2** (Non-stationary case). *When the offline data is non-stationary, i.e, the pdfs of $(H, S)$ and $(H', S')$ are different, we need to additionally require $\mathbb{E}[g_V(F') \mid S'] = V^{\pi^e}(S')$ in the definition.*

**Remark 3** (Comparisons with related works). *Similar concepts have been recently proposed in the context of confounded POMDPs [see e.g., SUHJ22, BK21]. However, our proposal significantly differs from theirs. First, their confounded setting does not cover our unconfounded setting because their behavior policies $\pi^b$ is* not *allowed to depend on current observations as mentioned in Section 1.1. Secondly, their proposal heir proposal overlooks the significant aspect of incorporating multi-step future observations, which plays a pivotal role in facilitating the existence of future-dependent value functions, as will be discussed in Example 1. For a detailed discussion, refer to Section B.*

---

**Algorithm 1** Minimax OPE on POMDPs

---

**Require:** Dataset $\mathcal{D}$, function classes $\mathcal{Q} \subset [\mathcal{F} \to \mathbb{R}], \Xi \subset [\mathcal{H} \to \mathbb{R}]$, hyperparameter $\lambda \geq 0$

1: $\hat{b}_V = \mathrm{argmin}_{q \in \mathcal{Q}} \max_{\xi \in \Xi} \mathbb{E}_{\mathcal{D}_{\mathrm{tra}}}[\{\mu(A, O)\{R + \gamma q(F')\} - q(F)\}\xi(H) - \lambda \xi^2(H)]$.

2: **return** $\hat{J}_{\mathrm{VM}} = \mathbb{E}_{\mathcal{D}_{\mathrm{ini}}}[\hat{b}_V(f)]$

---

Finally, we present a theorem to identify the policy value.

**Theorem 1** (Identification)**.** *Suppose (1a) the existence of learnable future-dependent value functions (need not be unique); (1b) the invertiblity condition, i.e., any $g : \mathcal{S} \to \mathbb{R}$ that satisfies $\mathbb{E}[g(S) \mid H] = 0$ must also satisfy $g(S) = 0$ ( i.e., $g(s) = 0$ for almost every $s$ that belongs to the support of $S$), (1c) the overlap condition $w_{\pi^e}(S) := d_{\pi^e}(S)/P_{\pi^b}(S) < \infty, \mu(O, A) < \infty$. Then, for any $b_V \in \mathcal{B}_V$,*

$$J(\pi^e) = \mathbb{E}_{f \sim \nu_{\mathcal{F}}}[b_V(f)]. \tag{5}$$

We assume three key conditions: the observability condition (i.e., $\mathcal{B}_V \neq \emptyset$), the invertibility condition, and the overlap condition. We call Condition (1a) the observability condition since it is reduced to the well-known concept of observability in the LQG control theory, as we will see in Section D. While Condition (1a) itself is concerned with learnable future-dependent value functions, it is implied by the existence of *unlearnable* future-dependent value functions according to Lemma 1 which can be verified using Picard's theorem in functional analysis [CFR07]. In general, the observability condition requires the future proxy $F$ to contain sufficient information about $S$ and is likely to hold when $F$ consists of enough future observations. We will see more interpretable conditions in the tabular POMDPs (Example 1) and POMDPs with Hilbert space embeddings (HSE-POMDPs) where the underlying dynamical systems have linear conditional mean embeddings (Example 5 in Section D).

The invertibility condition is imposed to ensure that a learnable future-dependent value function $b_V$ satisfies (3) (note that right hand side of Eq. 3 is a function of $Z, S$ instead of $H$). Again, we will present more interpretable conditions in the tabular and linear settings below and in Section D. Roughly speaking, it requires $H$ to retain sufficient information about $S$. In that sense, the history proxy serves as an instrumental variable (IV), which is widely used in economics [Hor11, New13] [5].

Finally, the overlap condition is a standard assumption in OPE [UIJ$^+$21].

**Example 1** (Tabular Setting)**.** *In the tabular case, abstract assumptions in Theorem 1 are reduced to certain rank assumptions. We first define $\mathcal{S}_b = \{s \in \mathcal{S} : P_{\pi^b}(s) > 0\}$. We define a matrix $\mathrm{Pr}_{\pi^b}(\mathbf{F} \mid \mathbf{S}_b) \in \mathbb{R}^{|\mathcal{F}| \times |\mathcal{S}|}$ whose $(i, j)$-th element is $\mathrm{Pr}_{\pi^b}(F = x_i \mid S = x'_j)$ where $x_i$ is the $i$th element in $\mathcal{F}$, and $x'_j$ is the $j$th element in $\mathcal{S}_b$. We similarly define another matrix $\mathrm{Pr}_{\pi^b}(\mathbf{S}_b, \mathbf{H})$ whose $(i, j)$-th element is $\mathrm{Pr}_{\pi^b}(S = x'_i, H = x''_j)$ where $x''_j$ denotes the $j$th element in $\mathbf{H}$.*

**Lemma 2** (Sufficient conditions for observability and invertibility )**.** *(a) When $\mathrm{rank}(\mathrm{Pr}_{\pi^b}(\mathbf{F} \mid \mathbf{S}_b)) = |\mathcal{S}_b|$, future-dependent value functions exist. Then, from Lemma 1, learnable future-dependent value functions exist. (b) The invertiblity is satisfied when $\mathrm{rank}(\mathrm{Pr}_{\pi^b}(\mathbf{S}_b, \mathbf{H})) = |\mathcal{S}_b|$.*

*The first two conditions require that the cardinalities of $\mathcal{F}$ and $\mathcal{H}$ must be greater than or equal to $\mathcal{S}_b$, respectively. The proof of Lemma 2 is straightforward, as integral equations reduce to matrix algebra in the tabular setting. We note that similar conditions have been assumed in the literature on HMMs and POMDPs [SBS$^+$10, BSG11, BGG13]. In particular, $\mathrm{Pr}_{\pi^b}(\mathbf{O} \mid \mathbf{S}_b) = |\mathcal{S}_b|$ has been imposed in previous works [NJ21, SUHJ22] in the context of confounded POMDPs. Nonetheless, our condition $\mathrm{Pr}_{\pi^b}(\mathbf{F} \mid \mathbf{S}_b) = |\mathcal{S}_b|$ is strictly weaker when $\mathcal{F}$ includes multi-step future observations, demonstrating the advantage of incorporating multi-step future observations compared to utilizing only the current observation. A more detailed discussion can be found in Appendix D.1.*

## 4 Estimation with General Function Approximation

In this section, we demonstrate how to estimate the value of a policy based on the results presented in Section 3. We begin by outlining the proposed approach for estimating $b_V(\cdot)$. The key observation is

---

[5]The observation that history can serve as an instrumental variable in POMDPs is mentioned in [HDG15, VSH$^+$16]. However, these works aim to learn the system dynamics instead of policy evaluation or learning. Hence, concepts like future-dependent value functions do not appear in these works.

that it satisfies $\mathbb{E}[L(b_V, \xi)] = 0$ for any $\xi : \mathcal{H} \to \mathbb{R}$ where $L(q, \xi)$ is defined as

$$L(q, \xi) := \{\mu(A, O)\{R + \gamma q(F')\} - q(F)\}\xi(H)$$

for $q : \mathcal{F} \to \mathbb{R}$ and $\xi : \mathcal{H} \to \mathbb{R}$. Given some constrained function classes $\mathcal{Q} \subset [\mathcal{F} \to \mathbb{R}]$ and $\Xi \subset [\mathcal{H} \to \mathbb{R}]$ and a hyperparameter $\lambda \geq 0$, the estimator is computed according to Line 1 of Algorithm 1. When the realizability $\mathcal{B}_V \cap \mathcal{Q} \neq \emptyset$ holds and $\Xi$ is unconstrained, we can easily show that the population-level minimizers $\text{argmin}_{q \in \mathcal{Q}} \max_{\xi \in \Xi} \mathbb{E}[L(q, \xi) - 0.5\lambda\xi^2(H)]$ are all learnable future-dependent value functions. This is later formalized in Theorem 2.

We can use any function classes such as neural networks, RKHS, and random forests to parameterize $\mathcal{Q}$ and $\Xi$. Here, the function class $\Xi$ plays a critical role in measuring how $q$ deviates from the ground truth $b_V$. The hyperparameter $\lambda$ is introduced to obtain a fast convergence rate. We call it a stabilizer instead of a regularizer since $\lambda$ does not need to shrink to zero as $n$ approaches infinity. Note that regularizers are needed when we penalize the norms of $q$ and $\xi$, i.e.,

$$\hat{b}_V = \underset{q \in \mathcal{Q}}{\text{argmin}} \max_{\xi \in \Xi} \mathbb{E}_{\mathcal{D}}[L(q, \xi) - 0.5\lambda\xi^2(H)] + 0.5\alpha'\|q\|_{\mathcal{Q}}^2 - 0.5\alpha\|\xi\|_{\Xi}^2,$$

for certain function norms $\|\cdot\|_{\mathcal{Q}}$ and $\|\cdot\|_{\Xi}$ and hyperparameters $\alpha', \alpha > 0$.

In the remainder of this section, we present three concrete examples using linear models, RKHSs and neural networks. Let $\|\cdot\|_{\mathcal{Q}}$ and $\|\cdot\|_{\Xi}$ denote $L^2$-norms when we use linear models and RKHS norms when using RKHSs.

**Example 2** (Linear models). *Suppose $\mathcal{Q}, \Xi$ are linear, i.e., $\mathcal{G} = \{\theta^\top \phi_{\mathcal{F}}(\cdot) \mid \theta \in \mathbb{R}^{d_{\mathcal{F}}}\}, \Xi = \{\theta^\top \phi_{\mathcal{H}}(\cdot) \mid \theta \in \mathbb{R}^{d_{\mathcal{H}}}\}$ with features $\phi_{\mathcal{F}} : \mathcal{F} \to \mathbb{R}^{d_{\mathcal{F}}}, \phi_{\mathcal{H}} : \mathcal{H} \to \mathbb{R}^{d_{\mathcal{H}}}$. Then,*

$$\hat{b}_V(\cdot) = \phi_{\mathcal{F}}^\top(\cdot) \left\{ \mathbf{M}_2^\top \{\alpha I_{d_{\mathcal{H}}} + \lambda \mathbf{M}_3\}^{-1} \mathbf{M}_2 + \alpha' I_{d_{\mathcal{F}}} \right\}^{-1} \mathbf{M}_2^\top \{\alpha I_{d_{\mathcal{H}}} + \lambda \mathbf{M}_3\}^{-1} \mathbf{M}_1,$$

$\mathbf{M}_1 = \mathbb{E}_{\mathcal{D}}[\mu(O, A)R\phi_{\mathcal{H}}(H)], \mathbf{M}_2 = \mathbb{E}_{\mathcal{D}}[\phi_{\mathcal{H}}(H)\{\phi_{\mathcal{F}}^\top(F) - \gamma\mu(O, A)\phi_{\mathcal{F}}^\top(F')\}], \mathbf{M}_3 = \mathbb{E}_{\mathcal{D}}[\phi_{\mathcal{H}}(H)\phi_{\mathcal{H}}^\top(H)].$ *When $\alpha' = 0, \alpha = 0, \lambda = 0$, the value estimators boil down to*

$$\hat{b}_V(\cdot) = \phi_{\mathcal{F}}^\top(\cdot)\mathbf{M}_2^+\mathbf{M}_1, \quad \hat{J}_{\text{VM}} = \mathbb{E}_{f \sim \nu_{\mathcal{F}}}[\phi_{\mathcal{F}}^\top(f)]\mathbf{M}_2^+\mathbf{M}_1.$$

*The above estimators are closely related to the LSTD estimators in MDPs [LP03]. Specifically, the off-policy LSTD estimator for state-value functions [DNP$^+$14, UHJ20] is given by*

$$\mathbb{E}_{s \sim \nu_{\mathcal{S}}}[\phi_{\mathcal{S}}^\top(s)]\mathbb{E}_{\mathcal{D}}[\phi_{\mathcal{S}}(S)\{\phi_{\mathcal{S}}^\top(S) - \gamma\mu(S, A)\phi_{\mathcal{S}}^\top(S')\}]^+\mathbb{E}_{\mathcal{D}}[\mu(S, A)R\phi_{\mathcal{S}}(S)]. \tag{6}$$

*Our new proposed estimator $\hat{J}_{VM}$ is*

$$\mathbb{E}_{f \sim \nu_{\mathcal{F}}}[\phi_{\mathcal{F}}^\top(f)]\mathbb{E}_{\mathcal{D}}[\phi_{\mathcal{H}}(H)\{\phi_{\mathcal{F}}^\top(F) - \gamma\mu(O, A)\phi_{\mathcal{F}}^\top(F')\}]^+\mathbb{E}_{\mathcal{D}}[\mu(O, A)R\phi_{\mathcal{H}}(H)],$$

*which is very similar to (6). The critical difference lies in that we use futures (including current observations) and histories as proxies to infer the latent state $S$ under partial observability.*

**Example 3** (RKHSs). *Let $\mathcal{Q}, \Xi$ be RKHSs with kernels $k_{\mathcal{F}}(\cdot, \cdot) : \mathcal{F} \times \mathcal{F} \to \mathbb{R}, k_{\mathcal{H}}(\cdot, \cdot) : \mathcal{H} \times \mathcal{H} \to \mathbb{R}$. Then,*

$$\hat{b}_V(\cdot) = \mathbf{k}_{\mathcal{F}}(\cdot)^\top \left\{ \{\mathbf{K}'_{\mathcal{F}}\}^\top \{\alpha I_n + \mathbf{K}_{\mathcal{H}}\}^{-1}\mathbf{K}'_{\mathcal{F}} + \alpha' I_n \right\}^{-1} \{\mathbf{K}'_{\mathcal{F}}\}^\top \mathbf{K}_{\mathcal{H}}^{1/2} \{\alpha I_n + \mathbf{K}_{\mathcal{H}}\}^{-1}\mathbf{K}_{\mathcal{H}}^{1/2}\mathbf{Y}$$

*where $\mathbf{Y} \in \mathbb{R}^n, k(\cdot) \in \mathbb{R}^n, \mathbf{K}_{\mathcal{H}} \in \mathbb{R}^{n \times n}, \mathbf{K}_{\mathcal{F}} \in \mathbb{R}^{n \times n}$ such that*

$$\{\mathbf{K}_{\mathcal{H}}\}_{(i,j)} = k_{\mathcal{H}}(H^{(i)}, H^{(j)}), \{\mathbf{K}_{\mathcal{F}}\}_{(i,j)} = k_{\mathcal{F}}(F^{(i)}, F^{(j)}), \{\mathbf{Y}\}_i = \mu(O^{(i)}, A^{(i)})R^{(i)},$$

$$\{\mathbf{K}'_{\mathcal{F}}\}_{(i,j)} = k_{\mathcal{F}}(F^{(i)}, F^{(j)}) - \gamma k_{\mathcal{F}}(F'^{(i)}, F^{(j)}), \{\mathbf{k}_{\mathcal{F}}(\cdot)\}_i = k_{\mathcal{F}}(\cdot, F^{(i)}).$$

**Example 4** (Neural Networks). *We set $\mathcal{Q}$ to a class of neural networks and recommend to set $\Xi$ to a linear or RKHS class so that the minimax optimization is reduced to single-stage optimization. The resulting optimization problem can be solved by off-the-shelf stochastic gradient descent (SGD) methods. Specifically, when we set $\Xi$ to a linear model, $\hat{b}_V$ is reduced to*

$$\hat{b}_V = \underset{q \in \mathbb{Q}}{\text{argmin}}\, Z(q)^\top \{\alpha I + \lambda \mathbb{E}_{\mathcal{D}}[\phi_{\mathcal{H}}(H)\phi_{\mathcal{H}}^\top(H)]\}^{-1} Z(q),$$

$$Z(q) := \mathbb{E}_{\mathcal{D}}[\mu(A, O)\{R + \gamma q(F')\} - q(F)\}\phi_{\mathcal{H}}(H)].$$

*When we set $\Xi$ to be an RKHS, $\hat{b}_V$ is reduced to*

$$\hat{b}_V = \underset{q \in \mathbb{Q}}{\text{argmin}}\{Z'(q)\}^\top \mathbf{K}_{\mathcal{H}}^{1/2}\{\alpha I_n + \lambda \mathbf{K}_{\mathcal{H}}\}^{-1}\mathbf{K}_{\mathcal{H}}^{1/2} Z(q),$$

$$\{Z'(q)\}_i := \mu(A^{(i)}, O^{(i)})\{R^{(i)} + \gamma q(F'^{(i)})\} - q(F^{(i)}).$$

**Remark 4** (Comparison with minimax methods for MDPs). *Our proposed methods are closely related to the minimax learning approach (a.k.a. Bellman residual minimization) for RL [ASM08]. More specifically, in MDPs where the evaluation policy is memory-less, [UIJ+21, TFL+20] proposed minimax learning methods based on the following equations,*

$$\mathbb{E}[h(S)\{\mu(S,A)\{\gamma V^{\pi^e}(S') + R\} - V^{\pi^e}(S)\}] = 0, \quad \forall h : \mathcal{S} \to \mathbb{R}.$$

*These methods are no longer applicable in POMDPs since we are unable to directly observe the latent state. Although substituting the latent state with the entire history can still provide statistical guarantees, it is susceptible to the curse of horizon.*

**Remark 5** (Modeling of system dynamics). *Our minimax estimators can be extended to learning system dynamics. In particular, for tabular POMDPs, these results are closely related to the literature on spectral learning in HMMs and POMDPs [SBS+10, BSG11, BGG13]. Refer to Section F,G.*

**Remark 6** (Finite horizon setting). *The extension to the finite horizon setting is discussed in Section E*

## 5  Finite Sample Results

We study the finite sample convergence rate of the proposed estimators in this section. To simplify the technical analysis, we impose three assumptions. First, we assume the function classes are bounded, i.e., $\|\mathcal{Q}\|_\infty \leq C_\mathcal{Q}, \|\Xi\|_\infty \leq C_\Xi$ for some $C_\mathcal{Q}, C_\Xi > 0$. Second, we assume that the offline data are i.i.d.[6] Third, we assume $\mathcal{Q}, \Xi$ are finite hypothesis classes. Meanwhile, our results can be extended to infinite hypothesis classes using the global/local Rademacher complexity theory; see e.g., [UIJ+21]. To simplify the presentation, following standard convention in OPE, we suppose the initial distribution is known, i.e., $|\mathcal{D}_{\text{ini}}| = \infty$.

**Accuracy of $\hat{b}_V$.**   We first demonstrate that $\hat{b}_V$ consistently estimates the learnable future-dependent value bridge functions. To formalize this, we introduce the following Bellman operators.

**Definition 3** (Bellman operators). *The Bellman residual operator onto the history is defined as*

$$\mathcal{T} : [\mathcal{F} \to \mathbb{R}] \ni q(\cdot) \mapsto \mathbb{E}[\mu(O,A)\{R + \gamma q(F')\} - q(F) \mid H = \cdot],$$

*and the Bellman residual error onto the history is defined as $\mathbb{E}[(\mathcal{T}q)^2(H)]$. Similarly, the Bellman residual operator onto the latent state, $\mathcal{T}^S$ is defined as*

$$\mathcal{T}^S : [\mathcal{F} \to \mathbb{R}] \ni q(\cdot) \mapsto \mathbb{E}[\mu(O,A)\{R + \gamma q(F')\} - q(F) \mid S = \cdot],$$

*and the Bellman residual error onto the latent state is defined as $\mathbb{E}[\{\mathcal{T}^S(q)\}^2(S)]$. The conditional expectations equal to zero when $h$ and $s$ lie outside the support of $H$ and $S$, respectively.*

The Bellman residual error onto the history is zero for any learnable future-dependent value function, i.e., $\mathbb{E}[(\mathcal{T}b_V)^2(H)] = 0$. Thus, this is a suitable measure to assess how well value function-based estimators approximate the true learnable future-dependent value functions.

**Theorem 2** (Finite sample property of $\hat{b}_V$). *Set $\lambda > 0$. Suppose (2a) $\mathcal{B}_V \cap \mathcal{Q} \neq 0$ (realizability) and (2b) $\mathcal{T}\mathcal{Q} \subset \Xi$ (Bellman completeness). With probability $1 - \delta$, we have $\mathbb{E}[(\mathcal{T}\hat{b}_V)^2(H)]^{1/2} \leq c\{1/\lambda + \lambda\} \max(1, C_\mathcal{Q}, C_\Xi)\sqrt{\frac{\ln(|\mathcal{Q}||\Xi|c/\delta)}{n}}$ where $c$ is some universal constant.*

Note (2a) and (2b) are commonly assumed in the literature on MDPs [CJ19, UIJ+21] as well. In particular, Bellman completeness means that the function class $\Xi$ is sufficiently rich to capture the Bellman update of functions in $\mathcal{Q}$.   For instance, these assumptions are naturally met in HSE-POMDPs. We require $\lambda > 0$ in the statement of Theorem 2 to obtain a parametric convergence rate. When $\lambda = 0$, although we can obtain a rate of $O_p(n^{-1/4})$, it is unclear whether we can achieve $O_p(n^{-1/2})$.

**Accuracy of $\hat{J}_{\text{VM}}$.**   We derive the finite sample guarantee for $\hat{J}_{\text{VM}}$.

---

[6]Without the independence assumption, similar results can be similarly established by imposing certain mixing conditions; see e.g., [SZLS20, LQM20, KU22].

**Theorem 3** (Finite sample property of $\hat{J}_{\mathrm{VM}}$)**.** *Set $\lambda > 0$. Suppose (2a), (2b), (2c) any element in $q \in \mathcal{Q}$ that satisfies $\mathbb{E}[\{\mathcal{T}^S(q)\}(S) \mid H] = 0$ also satisfies $\mathcal{T}^S(q)(S) = 0$. (2d) the overlap $\mu(O, A) < \infty$ and any element in $q \in \mathcal{Q}$ that satisfies $\mathcal{T}^S(q)(S) = 0$ also satisfies $\mathcal{T}^S(q)(S^\diamond) = 0$ where $S^\diamond \sim d_{\pi^e}(s)$. With probability $1 - \delta$, we have*

$$|J(\pi^e) - \hat{J}_{\mathrm{VM}}| \leq \frac{c(1/\lambda + \lambda)}{(1-\gamma)^2} \max(1, C_\mathcal{Q}, C_\Xi) \mathrm{IV}_1(\mathcal{Q}) \mathrm{Dr}_\mathcal{Q}[d_{\pi^e}, P_{\pi^b}] \sqrt{\frac{\ln(|\mathcal{Q}||\Xi|c/\delta)}{n}}, \quad (7)$$

*where*

$$\mathrm{IV}_1^2(\mathcal{Q}) := \sup_{\{q \in \mathcal{Q}; \mathbb{E}[\{\mathcal{T}(q)(H)\}^2] \neq 0\}} \frac{\mathbb{E}[\{\mathcal{T}^S(q)(S)\}^2]}{\mathbb{E}[\{\mathcal{T}(q)(H)\}^2]}, \quad (8)$$

$$\mathrm{Dr}_\mathcal{Q}^2[d_{\pi^e}, P_{\pi^b}] := \sup_{\{q \in \mathcal{Q}; \mathbb{E}_{s \sim P_{\pi^b}}[\{\mathcal{T}^S(q)(s)\}^2] \neq 0\}} \frac{\mathbb{E}_{s \sim d_{\pi^e}}[\{\mathcal{T}^S(q)(s)\}^2]}{\mathbb{E}_{s \sim P_{\pi^b}}[\{\mathcal{T}^S(q)(s)\}^2]}. \quad (9)$$

On top of (2a) and (2b) that are assumed in Theorem 2, when $\mathrm{IV}_1(\mathcal{Q}) < \infty, \mathrm{Dr}_\mathcal{Q}(d_{\pi^e}, P_{\pi^b}) < \infty$, (2c) and (2d) hold, we have the non-vacuous PAC guarantee. The condition $\mathrm{Dr}_\mathcal{Q}(d_{\pi^e}, P_{\pi^b}) < \infty$ and (2d) are the overlap conditions, which are adaptive to the function class $\mathcal{Q}$ and are weaker than (1c). These are also used in MDPs [XCJ+21]. Here, $\mathrm{Dr}_\mathcal{Q}(d_{\pi^e}, P_{\pi^b})$ is a refined version of the density ratio and satisfies $\mathrm{Dr}_\mathcal{Q}(d_{\pi^e}, P_{\pi^b}) \leq w_{\pi^e}(S)$. The condition $\mathrm{IV}_1(\mathcal{Q}) < \infty$ and (2c) are characteristic conditions in POMDPs that quantify how much errors are properly translated from on $H$ to on $S$. Similar assumptions are frequently imposed in IV literature [DLMS20, CP12].

The upper error bound in (7) does not have explicit exponential dependence on the effective horizon $1/(1-\gamma)$. In particular, as shown in Section D, for tabular POMDPs and HSE-POMDPs, the terms $\mathrm{Dr}_\mathcal{Q}(d_{\pi^e}, P_{\pi^b})$ and $\kappa(\mathcal{Q})$ can be reduced to certain condition numbers associated with covariance matrices spanned by feature vectors; see (15) and (16) in Appendix D. Hence, unlike SIS-based methods, we are able to break the curse of horizon.

The numbers of future and history proxies included in $F$ and $H$ represents a tradeoff. Specifically, if $F$ contains enough observations, it is likely that (2a) will hold. Meanwhile, if $H$ contains enough observations, it is more likely that (2b) will hold. These facts demonstrate the benefits of including a sufficient number of observations in $F$ and $H$. However, the statistical complexities $\ln(|\mathcal{Q}||\Xi|)$ will increase with the number of observations in $F$ and $H$.

Lastly, it's worth noting that while our methods effectively address the curse of horizon, they may incur exponential growth concerning the number of future proxies used. This also applies to history proxies, which should be longer than the length of short memory policies. Here, we focus on the explanation of future proxies. For instance, in the tabular case, $\log |\Omega|$ might scale with $|\mathcal{O}|^{M_F}$ when considering $\Omega$ as the set of all functions on $\mathcal{O}^{M_F}$. However, this situation differs significantly from the curse of horizon, a challenge that naive methods like replacing states with the entire history encounter. These methods would necessitate the entire history to achieve Markovianity, whereas we only require a shorter length of future observations to establish the conditions outlined in Theorem 1 (existence), which can be much shorter. Specific examples are provided throughout the paper, including Example 1, which discusses the tabular setting and demonstrates that we essentially need as many future proxies as states, as long as there is sufficient statistical dependence between them.

## 6 Experiment

This section empirically evaluates the performance of the proposed method on a synthetic dataset.[7]

We use the CartPole environment provided by OpenAI Gym [BCP+16], which is commonly employed in other OPE studies [SUHJ22, FCG18]. By default, this non-tabular environment consists of 4-dimensional states, which are fully observable. Following [SUHJ22], we create partial observability by adding independent Gaussian noises to each dimension of the state as $O^{(j)} = S^{(j)}(1 + \mathcal{N}(1 + 0.3^2)), 1 \leq j \leq 4$. To define behavior and evaluation policies, we first train an expert policy using DDQN [VHGS16] on latent states $S$. Subsequently, we apply Behavior Cloning (BC) on two datasets, one containing pairs of latent state and action $(S, A)$ and the other containing pairs of observation and action $(O, A)$, respectively. Then, we use the base policy obtained by BC on the state-action

---

[7]Our code is available at https://github.com/aiueola/neurips2023-future-dependent-ope

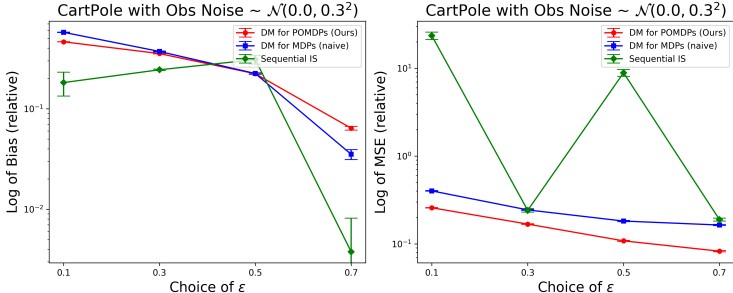

Figure 2: Logarithms of relative biases (left) and MSEs (right) of the proposed and the baseline estimators for various values of $\epsilon$, which specify the evaluation policy. The confidence intervals are obtained through 100 Monte Carlo simulations.

pairs $(S, A)$ to define an $\epsilon$-greedy behavior policy, where we set $\epsilon = 0.3$. [8] Similarly, the evaluation policy is also an $\epsilon$-greedy policy, based on the base policy obtained by BC on the observation-action pairs $(O, A)$, with different values of $\epsilon \in [0.1, 0.3, 0.5, 0.7]$. We conduct the experiment with 100 random seeds, and for each simulation, we collect logged data consisting of 1000 trajectories, each containing 100 steps.

We compare our proposal with Sequential Importance Sampling (SIS) [Pre00] and the naive minimax OPE [UHJ20], which is designed for fully-observable MDPs and does not account for partial observability. The naive minimax estimator is defined as if the environment was fully observable, replacing $H$ and $\bar{F}$ in Algorithm 1 with the current observation $O$. In contrast, our proposed method uses a 3-step history as $H$ and a one-step future as $F$ to address partial observability. Both our proposed method and the naive approach use two-layer neural networks for the function $\mathcal{Q}$ and RKHSs for $\Xi$, as detailed in Example 4.

We present the results in Figure 2, which demonstrate the superior accuracy of our proposed estimator compared to the baselines (SIS and naive minimax estimator) in terms of mean square errors (MSEs). Additional experimental details and ablation results, including the variations in the length of $H, \bar{F}$, and the choice of RKHSs, can be found in Appendix H.3.

## 7 Conclusion

We present a novel approach for OPE of short-memory policies in POMDPs. Our method involves introducing future-dependent value functions and the associated off-policy Bellman equations, followed by proposing a minimax estimator based on these equations. This is the first model-free method that allows for general function approximation and mitigates the curse of horizon. Our proposal is grounded in three interpretable key assumptions: observability, which asserts the presence of (short) future observations retaining adequate information about latent states, invertibility, which posits the existence of (short) histories preserving ample information about latent states; and the overlap between evaluation policies and behavior policies.

We have several avenues for enhancing our proposals. Firstly, automatically determining the appropriate lengths of futures and histories holds practical significance. Additionally, exploring recent attention mechanisms that extend beyond selecting the most recent history proxies or the nearest future proxies shows promise. Secondly, while we establish Bellman equations for POMDPs and use a simple minimax approach akin to [DLMS20], we may benefit from leveraging more refined methods introduced in recent research for solving conditional moment equations [BKM+23c, BKM+23a, BKM+23b].

## Acknowledgements

This material is based upon work supported by the National Science Foundation under Grant Nos. IIS 2112471, IIS 2141781, IIS 1846210, IIS 2154711 and an Engineering and Physical Sciences Research Council grant EP/W014971/1.

---

[8]When collecting offline data from the behavior policy, the behavior policy takes the observation $O$ as input, noting that the dimension of $S$ and $O$ is the same in our experimental setting.

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
