# OpenReview forum: "Future-Dependent Value-Based Off-Policy Evaluation in POMDPs"
_NeurIPS.cc/2023/Conference — NeurIPS 2023 spotlight_

### Official Review · Reviewer_4aE2 · 2023-07-03

**Soundness:** 3 good
**Presentation:** 4 excellent
**Contribution:** 3 good
**Rating:** 7
**Confidence:** 3

**Summary:**

- This paper proposes a novel future-dependent value function to study the problem of OPE in POMDPs with function approximations.
- A new Bellman-style conditional moment equation is derived for the future-dependent value function, based on which the value function can be learned via finite sample using minimax learning methods.
- Under certain completeness, realizability, and overlapping assumptions, a finite sample theoretical bound is derived for the proposed minimax learning algorithm. The paper also provides experimental evidence to empirically evaluate the performance of the algorithm.


**Strengths:**

- The paper first proposes using "future-dependent value function" to solve the problem of OPE in POMDPs, which is a critical problem in offline RL literature. The resulting method potentially overcomes the curse of horizon in learning POMDPs.
- The presentation of the paper is clear, e.g., using memory-less policy to illustrate the idea in the main text. The connections with highly related works, e.g., OPE in MDPs, OPE in confounded POMDPs, system dynamics learning, spectral learning, are well-discussed.
- The statistical theory regarding minimax learning and OPE is sound, and empirical evaluations are also provided.

**Weaknesses:**

- The idea of incorporating future information as parts of value functions for POMDPs is not new, e.g., [1], even though I still appreciate that this paper first attempts using such methods for OPE in POMDPs.
- The main part of the statistical theory regarding minimax estimation and OPE is standard given that various customized assumptions (completeness, realizability, and certain overlapping conditions) hold.

**References:**
[1] Uehara M, Sekhari A, Lee J D, et al. Provably efficient reinforcement learning in partially observable dynamical systems[J]. Advances in Neural Information Processing Systems, 2022, 35: 578-592.

**Questions:**

- Regarding the completeness assumption (2b), how can it be justified in the examples provided in the paper, e.g., HSE-POMDPs? I would appreciate it if the authors could provide more detailed clarifications on that.
- A novelty of the paper is using both history and future proxies in POMDP OPE. Is it possible to give a more quantitative trade-off between the usage of the history part and the future part (probably problem-dependent) towards optimal identifying and evaluating the policy value?
- Another seemingly key assumption in the presentation of the paper is that the offline data from an infinite horizon POMDP has a stationary distribution. Even mentioned in the main texts, I still wondering if the algorithm and theory can be directly extended to the more general non-stationary settings.

**Limitations:**

The above points in Questions are potential limitations.

---

> ### Author Rebuttal · Authors · 2023-08-08
>
> **Regarding the completeness assumption (2b), how can it be justified in the examples provided in the paper, e.g., HSE-POMDPs? I would appreciate it if the authors could provide more detailed clarifications on that.**
>
> Thank you for the question. Lemma 5 in Appendix D.2 establishes conditions for completeness under HSE-POMDPs. We will better advertise this result by remarking on it in the main text.
>
> **A novelty of the paper is using both history and future proxies in POMDP OPE. Is it possible to give a more quantitative trade-off between the usage of the history part and the future part (probably problem-dependent) towards optimal identifying and evaluating the policy value?**
>
> Thank you for bringing up this important point. As mentioned at the end of Page 8, we acknowledge the existence of a tradeoff, but at this stage, we do not have a concrete method to quantitatively characterize it. We will mention this as an important avenue for future work in our conclusion.
>
> **Another seemingly key assumption in the presentation of the paper is that the offline data from an infinite horizon POMDP has a stationary distribution. Even mentioned in the main texts, I still wondering if the algorithm and theory can be directly extended to the more general non-stationary settings.**
>
> We indeed assume that behavior policies are stationary, which is standard in the literature on POMDPs. However, the underlying distributions over latent states $S$ can be non-stationary. More specifically, from a methodological standpoint, it is possible to apply our methods even when the underlying state distributions are non-stationary. In such cases, adjustments to our current theoretical framework (Section 5) would be necessary. We believe that this could be achieved, for instance, by introducing mixing assumptions, as Shi et.al has done in the context of MDPs. We will mention this and explain the idea, but may not have the space to fully develop this direction in this paper.
>
> *Shi, C., Zhang, S., Lu, W., & Song, R. (2022). Statistical inference of the value function for reinforcement learning in infinite-horizon settings. Journal of the Royal Statistical Society Series B: Statistical Methodology, 84(3), 765-793.*
>
> Additionally, we want to highlight that our proposed approach can be readily extended to the finite horizon setting, and the details of this extension are given in Appendix E.

---

> > ### Comment · Reviewer_4aE2 · 2023-08-13
> > **Response to Authors**
> >
> > I have read the rebuttal and the comments from other reviewers. Thank you for the efforts dealing with my concerns and questions. I will keep my score as 7.

---

### Official Review · Reviewer_MVwZ · 2023-07-08

**Soundness:** 3 good
**Presentation:** 2 fair
**Contribution:** 3 good
**Rating:** 3
**Confidence:** 3

**Summary:**

The paper introduces a novel model-free approach for off-policy evaluation (OPE) in partially observable Markov decision processes (POMDPs). The authors propose the use of future-dependent value functions (FDVFs) to capture the latent states by leveraging future observations. They provide a Bellman-like equation and a minimax algorithm for FDVFs, along with a PAC analysis.

**Strengths:**

+ The paper addresses a crucial challenge of handling real-world data that are partially observable and difficult to collect.

+ The proofs seem to be correct.

**Weaknesses:**

- The readability of the paper can be enhanced, particularly by providing a more intuitive motivation and explanation of FDVFs. The paper is densely written with numerous references to other sections and papers, which disrupts the flow. It is important to clearly state the main assumptions, problem, and overall approach in an intuitive manner.

- The main weakness of the paper is the experiments section, which only considers a simple environment (CartPole) with partial observability achieved by adding Gaussian noise.

- Additionally, the proposed method utilizes a limited history (3-step) and one-step future, which may not sufficiently demonstrate its effectiveness in mitigating the curse of horizon.

- Furthermore, it is important to compare the proposed method with other OPE methods designed for POMDPs, as well as those using stacked observations.

**Questions:**

In Remark 3, "heir proposal" seems to be a typo; could you please clarify?

Instead of SIS, could SIS with resampling (a.k.a. SIR, SIS/R, SMC, particle filtering) be utilized instead?

---

> ### Author Rebuttal · Authors · 2023-08-08
>
> ### Rebuttal for the Weakness
>
> **The readability of the paper can be enhanced, particularly by providing a more intuitive motivation and explanation of FDVFs.....**
>
> Thank you for your suggestion. The primary motivation is FDVFs give embeddings of the latent states such that the expectation is the same as that of the latent-state value function, enabling OPE, as discussed in lines 172-174. To further help the reader we will move discussion of existence of FDVFs and examples from later in the text to be next to the definition.
>
> **The main weakness of the paper is the experiments section, which only considers a simple environment (CartPole) with partial observability achieved by adding Gaussian noise.**
>
> The offline policy evaluation problem is notoriously challenging, even in MDPs. It can be even more difficult than offline policy learning in the sense that we can still learn good policies without precise policy value estimation. Due to this difficulty, many representative papers focusing on offline policy evaluation use simple environments such as Cartpole, as exemplified in [TB16] and the following papers:
>
> Farajtabar, Mehrdad, Yinlam Chow, and Mohammad Ghavamzadeh. "More robust doubly robust off-policy evaluation." International Conference on Machine Learning. PMLR, 2018.
>
> Shi, Chengchun, Runzhe Wan, Victor Chernozhukov, and Rui Song. "Deeply-debiased off-policy interval estimation." In International conference on machine learning, pp. 9580-9591. PMLR, 2021.
>
> In our setting, we even consider POMDPs, which are even more challenging than MDPs. Therefore, we believe that Cartpole is a reasonably challenging environment for our paper, and it is not considered a toy environment for our task of offline policy evaluation in POMDPs, especially as our primary contributions are theoretical-methodological. But we of course want to make the paper even stronger and can easily add more experiments to support our results. We will therefore run our code also the Mountain car and taxi environments. Both of these are straightforward to do using the code base that we submitted as supplementary materials.
>
> **Additionally, the proposed method utilizes a limited history (3-step) and one-step future, which may not sufficiently demonstrate its effectiveness in mitigating the curse of horizon.**
>
> Our methods are designed to be flexible, allowing for varying numbers of history and future steps. We have included ablation studies to demonstrate the effects of changing these parameters in Appendix. We are confident that our approach effectively addresses the curse of horizon under interpretable and reasonable assumptions. Notably, our work stands as the sole work in the realm of OPE in POMDPs with a theoretical guarantee, which can avoid the curse of horizon.
>
> You might be interested in our choice of a 3-step history and a 1-step future. In the future, we plan to explore attention-based methods to determine which histories and futures we should select. If you require further clarification on specific concerns, we are more than willing to provide additional details.
>
> **Furthermore, it is important to compare the proposed method with other OPE methods designed for POMDPs, as well as those using stacked observations.**
>
> To the best of our knowledge, existing OPE methods for POMDPs in the non-tabular setting with statistical guarantees are limited to SIS. Therefore, we believe our comparison is reasonable. But there are indeed some practical heuristics we can compare to. We will therefore add the standard baseline that simply tackles the problem as an MDP after augmenting states with short histories, which we believe is what you are referring to. Since this will necessarily be biased it will not converge. Nonetheless we agree that plotting it will provide an illustration of the importance of statistical convergence guarantees.
>
> ### Rebuttal for the Questions
>
> **In Remark 3, "heir proposal" seems to be a typo; could you please clarify?**
>
> Many thanks for catching it! It will be corrected if our paper gets accepted.
>
> **Instead of SIS, could SIS with resampling (a.k.a. SIR, SIS/R, SMC, particle filtering) be utilized instead?**
>
> We acknowledge that in the literature of importance sampling, resampling is commonly used. However, in OPE, we specifically focus on scenarios where data is generated by certain policies, and we do not have control over the data-generating process. Consequently, implementing resampling in such cases may not be feasible. Nevertheless, we agree that the idea of resampling may be useful in the online setting, where control over the data-generating process is available. We will point to this possible direction as an avenue for future work on online learning in POMDPs in our conclusion.

---

> > ### Comment · Reviewer_MVwZ · 2023-08-14
> >
> > Thank you for your answers.
> > Currently, the paper is providing limited experimental evaluation to show effectiveness of the proposed approach,  comparing with existing methods -- while the authors' response suggest that additional tests would be run, at this point it is unclear how well the proposed method would work. Judging based on the results available, unfortunately I cannot raise the score.

---

> > > ### Author Response · Authors · 2023-08-18
> > >
> > > Thank you very much for your prompt response. In addition to additional experiments for extra illustration, we have sought to explain clearly why the proposed method would work. Please allow us to summarize the key points in response to your previously stated concerns. Please let us know if you have additional questions.
> > >
> > > ***Furthermore, it is important to compare the proposed method with other OPE methods designed for POMDPs, as well as those using stacked observations*.**
> > >
> > > As we explained in our response, there are **no "other OPE methods designed for POMDPs”**, i.e., OPE methods for POMDPs with statistical convergence guarantees, other than SIS, to which we do compare. Correspondingly, for any other alternative method, for $n$ large enough, their non-vanishing bias would dominate -- i.e., for large $n$ they would always do worse. That is clear without any additional experiments. If you know of other methods with guaranteed convergence, we would be grateful if you could kindly point it out to us.
> > >
> > > ***The main weakness of the paper is the experiments section, which only considers a simple environment (CartPole) with partial observability achieved by adding Gaussian noise.***
> > >
> > > As we argued in our response, CartPole is actually far from a simple environment in the setting we study (OPE in POMDPs), **as is supported by references to other representative works.** We would greatly appreciate it if you could suggest literature on OPE in POMDPs with statistical guarantees that investigate performance with more complex environments / partial observability
> > >
> > > ***Additionally, the proposed method utilizes a limited history (3-step) and one-step future, which may not sufficiently demonstrate its effectiveness in mitigating the curse of horizon.***
> > >
> > > Per our experiments, our method does demonstrate effective mitigation of the curse of horizon. **The only other method with guaranteed convergence is SIS, which is known to suffer from the curse of horizon.** Our improvement over it, both numerically in the experiment and theoretically in our guarantees, demonstrate how our method mitigates the curse. Additionally, we have an ablation study in Appendix H.5 in the supplemental materials to further study the effect of history and future length. If you remain unconvinced about our ability to effectively address the curse of horizon, we would greatly appreciate receiving more specific details or inquiries that we can address to further clarify the matter.
> > >
> > > **Additional points we would like to convey**
> > >
> > > We would definitely incorporate your suggestions into account to improve the presentation further.  But, we would like to emphasize that the primary contribution of our paper resides in the advancement of the **first** model-free OPE methods employing function approximation for POMDPs with statistical guarantees, effectively addressing the curse of the horizon. The experiments were conducted to showcase its utility, and it's noteworthy that our paper is significantly different from pure empirical papers that might demand lots of experiments.

---

### Official Review · Reviewer_RwWH · 2023-07-08

**Soundness:** 3 good
**Presentation:** 4 excellent
**Contribution:** 3 good
**Rating:** 7
**Confidence:** 4

**Summary:**

The paper considers off-policy evaluation for POMDPs, where the data is collected using a behavior policy that is only dependent on the observations available to the evaluation policy as well. The work defines a new "future-dependent" value function and its associated Bellman equations that are used to define a new set of conditional moment equations. While the existing estimators for this setting suffer from the curse of the horizon, the proposed method does not.  Overall, I think it is an interesting idea, though there are some points that can be clarified better.

**Strengths:**

- Interesting idea of using future-dependent value functions. Though some prior works have touch upon similar ideas, the proposal is different enough.

- OPE in POMDP is a common topic of interest and might be relevant to a lot of people.

**Weaknesses:**

- More discussions about Assumptions could be helpful

- Experiments can be improved

**Questions:**

1. Staring at Definition 1, I am wondering if the importance-weighted return $\rho G$ extracted from F, is a valid instance of $g_V(F)$? (I understand that propensity scores are not a part of $F$, but let's say data is augmented to make them part of $F$).

Complimentarily, some discussion + examples around Definition 1 about the existence of $g_V(F)$ would also be fruitful. Line 211 kind of gets at it, but wraps it up in technical jargon.

2. If yes, it is not intuitive to me why would the proposed method break the curse of the horizon.

Even if the answer to the above is no, I am wondering if there is a price being paid in terms of the "curse of the future", i.e., the length of $F$? As mentioned in remark 4, if address the problem at hand can be addressed by substituting states with the history observed so far (although $\mu$ would only depend on the current observation under the assumptions made earlier around Eqn 3 and 4.) and that is cursed by the length of the horizon, why wouldn't the proposed method be cursed by the length of the future?


3. When multiple $g_V$ functions exist that satisfy the required condition, is there any reason to prefer one over the other? Can additional regularization be added to bias the solution towards one of them?

4. Optional:  In my opinion, section 4 does not contribute much as once the conditional moment equations are obtained, the rest is a standard adaption of existing techniques.  I think it would be far more beneficial to a reader to have more discussion + positive/negative examples about the conditions for invertibility and existence, and what they imply in practice.


**Limitations:**

4. It would be beneficial to add another baseline, similar to naive minimax OPE, albeit with history as the state (with hyper-parameter tuning done similarly to the proposed method to choose the ideal history length).

Minor:

- Sentence phrasing in Line 129 suggests a model-based approach, whereas the main method is model-free
- Eqn after Line 135 has a typo in subscripts.
- Line 199: typo

---

> ### Author Rebuttal · Authors · 2023-08-08
>
> **Staring at Definition 1, I am wondering if the importance-weighted return $\rho G$ extracted from $F$, is a valid instance of $g_{V}(F)$?**
>
> You are right: this would work if $F$ were the infinite future including _all_ future observations, actions, and rewards. Then, this choice of $g_{V}(F)$ essentially leads to sequential importance sampling (SIS), which suffers the curse of horizon due to the size of importance weights $\rho$. To mitigate this, it becomes essential to use short futures with finite $M_F$ as we do. We will emphasize this point and the interesting connection to SIS.
>
> **Some discussion + examples around Definition 1 about the existence of $g_V(F)$ would also be fruitful. Line 211 kind of gets at it, but wraps it up in technical jargon.**
>
> Thanks for the great suggestion. We will add discussion right after Definition 1, explaining existence is a question of existence of a solution to an integral equation and give the example of discrete variables where it can easily be stated as a solution to a system of linear equations (where we do not need Picard's theorem).
>
> **If yes, it is not intuitive to me why would the proposed method break the curse of the horizon... why wouldn't the proposed method be cursed by the length of the future?**
>
> Your observation is indeed astute. When employing multi-step futures, without a specific inductive bias (e.g., linear or smooth), the error could potentially grow exponentially in the number of future proxies used. For instance, in the tabular case, $\log |\Omega|$ might scale with $|O|^{M_F}$ when considering $\Omega$ as the set of all functions on $|O|^{M_F}$. However, this scenario is notably different from the curse of horizon, which is a challenge that naive methods (such as replacing states with the entire history) face. Whereas such methods would suffer the length of the history that is needed to induce Markovianity (in general, the _whole_ history), we would only suffer the length $M_F$ of the future that is needed to establish the conditions of Theorem 1 (existence and invertibility) which can require much shorter length. We give specific examples throughout the paper. Example 1 in the main text discusses the tabular setting, showing we essentially need as many futures as the state, as long as there is sufficient statistical dependence between them. In Appendix D.2 we give the example of HSE-POMDPs and LQGs and show how they satisfy these conditions. We will add this discussion and make clearer in what ways do in fact break the curse of horizon.
>
> To summarize:
>
> * The curse of horizon is unavoidable in naive methods, as you mentioned, where the state is replaced with the entire history in POMDPs. While using a short history instead of the whole history may be an option, it either requires a high-order Markov assumption or it may not provide any guarantees. In contrast, in our proposal, by choosing the length $M_F$ appropriately, we can ensure statistical guarantees for any horizon, provided the futures satisfy our proxy conditions.
>
> * The length of $M_F$ may indeed be relatively short in order to satisfy our conditions. For instance, in the tabular case, our assumption aligns with those made in several POMDP papers [ZUSL22, NJ21, SUHJ22], as mentioned in Line 234 and Line 235. Since these papers also encounter errors that grow exponentially with the length of future-proxies, they too implicitly assume a short length for future-proxies
>
> * The curse of horizon is inevitable without any assumptions, as demonstrated in [JKKL20]. To address this challenge, we propose new methods based on three assumptions: (a) (short) future observations serve as proxies for latent states, (b) histories serve as proxies for latent states, and (c) the length of the memory is short. We consider these assumptions to be interpretable and reasonable.
>
> **When multiple $g_V$ functions exist that satisfy the required condition, is there any reason to prefer one over the other? Can additional regularization be added to bias the solution towards one of them?**
>
> Thank you for the excellent question. We will add a discussion of this. In the present paper, we focus on estimation with finite-sample bounds, for which any $b_V$ would do. (Note $b_V$ are the value functions we learn, while $g_V$ are fundamentally unlearnable.) Any $b_V$ satisfying $(\mathcal Tb_V)(H)=0$ is sufficient for identification. And, in our estimation, we only establish guarantees on $\mathbb E[(\mathcal T\hat b_V)^2(H)]$ in Theorem 2, which suffice to establish bounds for evaluation in Theorem 3. This can be interpreted as bounds on the projected distance $\mathbb E[\mathcal T(\hat b_V-b_V)^2(H)]$, which is the _same_ no matter _which_ $b_V$ we choose.
>
> A future direction may be to consider inference based on asymptotic normality. To address that goal, it may be essential to include explicit regularization, such as Tikhonov regularization, as has been done in a different but related setting:
>
> *Bennett, Andrew, Nathan Kallus, Xiaojie Mao, Whitney Newey, Vasilis Syrgkanis, and Masatoshi Uehara. "Source Condition Double Robust Inference on Functionals of Inverse Problems." arXiv preprint arXiv:2307.13793 (2023).*
>
> **Optional: In my opinion, section 4 does not contribute much as once the conditional moment equations are obtained, the rest is a standard adaption of existing techniques. I think it would be far more beneficial to a reader to have more discussion + positive/negative examples about the conditions for invertibility and existence, and what they imply in practice.**
>
> We understand that an expert reader, such as yourself, may find it easy to derive the equations in Section 4 after reading Section 3. However, we also acknowledge that for many readers, this part may not be so trivial. Therefore, we think it best to retain it. Nevertheless, we agree with you that we should provide more discussion on invertibility and existence, and we will incorporate these discussions, as detailed above.

---

### Official Review · Reviewer_489Z · 2023-07-12

**Soundness:** 3 good
**Presentation:** 3 good
**Contribution:** 3 good
**Rating:** 7
**Confidence:** 4

**Summary:**

The paper addresses the problem of off-policy evaluation in POMDPs. The main idea is to modify the Bellman equation by incorporating future and historical observations instead of relying on unobserved states. This new future-dependent value function allows for policy evaluation without the need for unobserved states. The paper explores the sufficient conditions and conducts finite sample analysis based on the modified Bellman equation. The proposed algorithm is demonstrated through a numerical experiment on CartPole, showing improved prediction error.

**Strengths:**

Motivation and clarity:
- The paper acknowledges the common occurrence of partial observability in real-world scenarios and the challenges it poses for off-policy evaluation, making it a relevant and long-standing research problem.
- The paper is well-written and easy to follow.

Novelty:
- The proposed methods are clever and remarkably simple.
- The discussion on the conditions for observability and invertibility provides theoretical insights into the occurrence of "magic" (evaluation without observing unobserved states). The entire reasoning process almost feels like a result from a standard RL textbook.

**Weaknesses:**

My only concern is the practicality of the proposed algorithm. The experimental results are not sufficiently convincing, and conducting more extensive studies, including (1) more baselines and (2) more complex environments, particularly real-world POMDP applications, would greatly enhance the paper.

**Questions:**

Questions:
1. The definition of $\mathcal{F}$ for future observations is not consistent. In the equation after line 135 (also Figure 1), $F$ consists of $M_F$ observations and $M_F-1$ actions. However, in lines 142-147, $F$ consists of $M_F$ observations and $M_F$ actions. Which one is actually used in the experiment?
2. (Question regarding the experiment, after reviewing Appendix H.3)

Suggestions and Typos
1. Typo: In the equation after line 135, $(O_{-M_H:t-1}, A_{-M_H:t-1})$ should be $(O_{t-M_H:t-1}, A_{t-M_H:t-1})$.
2. Typo: In line 199, remove "heir proposal."
3. It would be interesting for practitioners to observe the prediction error as $M_F$ and $M_H$ vary. Exploring the trade-off between more accurate prediction through increased history/future observations and the complexity of the function space would be an intriguing avenue for future research.
4. An extension to attention-based neural networks as the function space would be extremely interesting.

---

> ### Author Rebuttal · Authors · 2023-08-08
>
> **My only concern is the practicality of the proposed algorithm. The experimental results are not sufficiently convincing, and conducting more extensive studies, including (1) more baselines and (2) more complex environments, particularly real-world POMDP applications, would greatly enhance the paper.**
>
>
> Thank you for the suggestion. We will incorporate additional experiments to answer questions of practicality. Our primary contributions are theoretical-methodological, and we hope you agree the paper is acceptable in its current form given this focus. But we of course want to make the paper even stronger and can easily add more experiments to support our results. Regarding (1), we will add the standard baseline that simply tackles the problem as an MDP after augmenting states with short histories. Regarding (2), we will run our code the Mountain car and taxi environments. Both of these are straightforward to do using the code base that we submitted as supplementary materials.
>
>
>
> **The definition of F for future observations is not consistent**
>
> Thank you for bringing this to our attention. Our proposal actually works in both cases: whether we have $M_F$ or $M_F-1$ actions. However, for consistency, we generally consider the case where future observations consist of $M_F$ observations and $M_F-1$ actions. We will fix lines 142-147 to be consistent with this choice, and we will mention explicitly in a remark that the alternative option of $M_F$ actions also works.
>
> **Possible extensions.**
>
> We appreciate the reviewer's valuable suggestions. We agree that both extensions are indeed very interesting avenues for future research. We will certainly mention and explore these directions in the conclusion. Especially using attention mechanisms could potentially provide a very practical way to select relevant histories and future observations.

---

### Official Review · Reviewer_W56S · 2023-07-22

**Soundness:** 3 good
**Presentation:** 3 good
**Contribution:** 4 excellent
**Rating:** 7
**Confidence:** 3

**Summary:**

The paper proposes a new OPE method for in POMDPs. The paper first introduces a new future-dependent value function and derive a new Bellman equation for the future-dependent value functions. Then, the paper proposes an estimator to learn such future-dependent value function. The paper provides theoretical results on the finite sample guarantee and a simple empirical demonstration that the estimator performs well in a partial observable Cartpole environment.

**Strengths:**

The paper introduces a new estimator for POMDPs, which is claimed to be the first model-free method that enable general function approximation and does not suffer from the curse of horizon. I think the paper would have a significant contribution to the OPE community.
(Note that I am not familiar with the literature on OPE for POMDPs, so my evaluation is sorely based on the related work section in the paper).

The future dependent idea is new, as far as I know. The paper also sufficiently discuss related works on OPE for POMDPs and the corresponding minimax estimator for fully observable MDPs. It is also good that the paper provides many examples on when the conditions hold in different settings such as tabular setting, and the connection between these conditions to conditions in existing works.

I didn’t check the proof in the appendix carefully, but the theoretical results are reasonable.

**Weaknesses:**

I think one weakness of the paper is the presentation clarity. It introduce many notation so it requires careful reading at first. I understand that these notations are necessary but I think some improvement can be made to make the paper easier to read.  Some steps/derivations are not very clear to me:
- Are all expectation in the paper, when not explicitly specified, taken with respect to the offline data sampled from the data collection policy $\pi_b$?
- The expectation in definition 1 is taken w.r.t. the data from the data collection policy? What is the intuition behind the definition? If we can sample history from the data collection policy, there exists some function that can map the history to the true return?
- In Line 218, “note that right hand side of Eq. 3 is a function of Z,S instead of H”. What is Z here?
- The paper is about OPE for short-memory policies, but the main text only describes methods for memory-less policies (for simplicity). I think the paper can still  briefly summarizing what is different for short-memory policies. Does the length of the memory affect the finite sample bounds?
- The paper mentions that it addresses the curse of horizon, however, isn't this because (1) we only consider short-memory policies and (2) assume future F contains sufficient information about S?

In the experiment, the base policy (I guess it is the data collection policy?) is defined on (S,A) but I think the method requires $\pi_b(A|O)$. Did I misunderstand something here?


**Questions:**

I would appreciate it if the authors can answer my questions listed in the weakness section, especially on the curse of horizon.

**Limitations:**

The authors adequately addressed the limitations. They provide many extension of the proposed method in different settings in the appendix (e.g. for finite horizon).

---

> ### Author Rebuttal · Authors · 2023-08-08
>
> Thank you for your incisive comments. We will review the draft again, and improve the clariy.
>
>
> **Are all expectations in the paper, when not explicitly specified, taken with respect to the offline data sampled from the data collection policy?**
>
> Yes. We will make it more explicit and reiterate relevant equations.
>
>
> **The expectation in definition 1 is taken w.r.t. the data from the data collection policy?**
>
> Yes.
>
> **Intuition behind the definition 1? Why don't we use the histories?**
>
> This is an excellent comment! We first explain the intuition. The intuition is to embed the original value functions, defined on the latent states, into ones that take multi-step futures as inputs. This allows us to derive a new Bellman equation for these future-dependent value functions.
>
> Next, we discuss why we chose to use multi-step futures instead of multi-step histories. The reason behind this choice is that if we were to use multi-step histories as inputs instead of multi-step futures, it remains unclear how to estimate such new value functions. Specifically, the Bellman equation might not hold if we replace the latent state with the multi-step histories.
>
>
> **In Line 218, “note that right hand side of Eq. 3 is a function of Z,S instead of H”. What is Z here?**
>
> Thank you for catching the typo. It should just say "of S". We will fix it accordingly.
>
>
> **I think the paper can still briefly summarizing what is different for short-memory policies. Does the length of the memory affect the finite sample bounds?**
>
> This is another excellent comment! To save space, we discussed the evaluation of short-memory policies in Section C. Following your suggestion, we will provide a summary in the main text if our paper gets accepted. In particular, The length of memory indeed affects the bound, and this would be reflected in $\log |\mathcal{Q}|$, as the input of the future-dependent value function depends on the length of the memory. Without inductive bias, in the worst case, this error could grow exponentially fast with respect to the length of memory. Notice that this differs from the curse of horizon suffered by naive methods such as SIS. The curse of horizon exists even when the evaluation policy is memoryless.
>
> **The paper mentions that it addresses the curse of horizon, however, isn't this because (1) we only consider short-memory policies and (2) assume future F contains sufficient information about S?**
>
> Yes, you are right. Both are our primary assumptions. Indeed, without any assumption, it is unclear how to break the curse of horizon. Many papers formally proved that the curse of horizon is unavoidable without any assumptions in POMDPs [see e.g., JKKL20].
>
> We believe that our assumptions are reasonable for the purpose of overcoming the curse of horizon. Specifically, short-memory policies are widely used in practice to handle non-Markov environments, as mentioned in the introduction (E.g., in DQN in the Atari [MKS+ 13], the length is $4$). Additionally, similar assumptions regarding the future are widely imposed in the POMDP literature; see e.g., [JKKL20, NJ21, SUHJ22].
>
>
> **In the experiment, the base policy (I guess it is the data collection policy?) is defined on (S,A) but I think the method requires ... Did I misunderstand something here?**
>
> Apologize for the confusion. This is a typo. In line 344, $(S,A)$ should be $(O,A)$. We will fix it accordingly. Thanks for catching it.

---

> > ### Comment · Reviewer_W56S · 2023-08-11
> > **Response to Authors**
> >
> > Thank you for the detailed answers. I’ve read the author response and other reviews and I will keep my evaluation and rating.

---

### Decision · Program_Chairs · 2023-09-21

**Decision:**

Accept (spotlight)

**Comment:**

The paper addresses off-policy evaluation in POMDPs and introduces a novel future-dependent value function. Instead of relying on unobserved states, this approach modifies the Bellman equation to incorporate both future and historical observations. The paper derives new Bellman and conditional moment equations and proposes an estimator that circumvents the curse of the horizon issue faced by existing methods. Theoretical results on the finite sample guarantee are presented. This paper is highly impactful for the community it is addressing. Overall a very solid submission, a clear accept.